# Diversity and saline-alkali resistance of Coleoptera endosymbiont bacteria in arid and semi-arid climate

Haitao Yue,[1,2] Xiaoyun Ma,[1] Shuwen Sun,[1] Hongying Hu,[1] Jieyi Wu,[1] Tong Xu,[1] Danyang Huang,[1] Yiqian Luo,[1] Junqiang Wu,[1] Tingting Huang[1]

**ABSTRACT** Soil salinization usually occurs in arid and semi-arid climate areas from 37 to 50 degrees north latitude and 73 to 123 degrees east longitude. These regions are inhabited by a large number of Coleopteran insects, which play an important role in the ecological cycle. However, little is known about the endosymbiotic microbial taxa and their biological characteristics in these insects. A study of endosymbiotic microorganisms of Coleoptera from Xinjiang, a typical arid and inland saline area, revealed that endosymbiont bacteria with salinity tolerance are common among the endosymbionts of Coleoptera. Functional prediction of the microbiota analysis indicated a higher abundance of inorganic ion transporters and metabolism in these endosymbiont strains. Screening was conducted on the tolerable 11% NaCl levels of *Brevibacterium casei* G20 (PRJNA754761), and differential metabolite and proteins were performed. The differential metabolites of the strain during the exponential and plateau phases were found to include benzene compounds, organic acids, and their derivatives. These results suggest that the endosymbiotic microorganisms of Coleoptera in this environment have adaptive evolution to extreme environments, and this group of microorganisms is also one of the important resources for mining saline and alkaline-tolerant chassis microorganisms and high-robustness enzymes.

**IMPORTANCE** Coleoptera insects, as the first largest order of insect class, have the characteristics of a wide variety and wide distribution. The arid and semi-arid climate makes it more adaptable. By studying the endosymbiont bacteria of Coleoptera insects, we can systematically understand the adaptability of endosymbiont bacteria to host and special environment. Through the analysis of endosymbiont bacteria of Coleoptera insects in different saline-alkali areas in arid and semi-arid regions of Xinjiang, it was found that bacteria in different host samples were resistant to saline-alkali stress. These results suggest that bacteria and their hosts co-evolved in response to this climate. Therefore, this study is of great significance for understanding the endosymbiont bacteria of Coleoptera insects and obtaining extremophile resources (Saline-alkali-resistant chassis strains with modification potential for the production of bulk chemicals and highly robust industrial enzymes).

**KEYWORDS** arid and semi-arid climate, saline-alkali resistance, Coleoptera, endosymbiont bacteria, proteomics, metabolomics

In arid and semi-arid areas, high temperatures and low rainfall are important factors contributing to soil salinization (1). Due to its unique geographical position, China is classified as one of the seven major dryland countries in Asia (2). Xinjiang, located in northwest China, is far away from the sea and experiences a typical temperate continental arid and semi-arid climate. The prominent feature of this area is the combination of low precipitation and high evaporation, which leads to the enrichment of salt on the soil

Address correspondence to Haitao Yue, yuehaitao@tsinghua.org.cn.

Haitao Yue and Xiaoyun Ma contributed equally to this article. Author order was determined based on ther contribution to the article.

The authors declare no conflict of interest.

See the funding table on p. 18.

surface as water evaporates, resulting in severe soil salinization (3). Therefore, while some studies focus on soil improvement and the exploration of microbial resources within the soil, there is a relative lack of research on the diversity of species inhabiting the saline-alkali regions at all levels (4).

Coleoptera, the largest order of insects, exhibits a wide variety of species and distribution that can adapt to climate conditions in most regions of the world, including arid and semi-arid climates (5, 6). Despite the lower species diversity in arid regions compared to humid regions (7), many insects under the Coleoptera not only survive but also occupy significant ecological positions in these regions. Coleoptera plays a crucial role in maintaining ecological balance in arid and semi-arid climates by providing food sources for carnivorous animals and pollinating plants (8). The diet of Coleoptera insects in arid areas is complex, with phytophagous species often consuming crops (9), carnivorous species preying on plant-eating arthropods and mollusks (10), and saprophytic species feeding on animal and plant remains or excrement, participating in the natural energy and material cycle of Scarabaeinae (11, 12). Given their living environment of drought and salinity, important ecological status, and complex feeding habits, exploring the composition and function of endosymbiont bacteria of Coleoptera insects living in arid and semi-arid areas is crucial. It can verify whether Coleoptera insects can be used as a mining route for saline-tolerant microorganisms and also provide insights into the composition and function of endosymbionts of different Coleoptera insects living in this climate region after long-term symbiosis and co-evolution with their hosts.

Saline-tolerant microorganisms have been widely used in the microbial industry due to their tolerance to saline-alkali environment (13, 14). In the current study, the researchers primarily sourced these microorganisms from natural environments that are rich in salt and alkali (15, 16). They found that similar screening habitats resulted in similar tolerance mechanisms and metabolic synthesis abilities among the isolated microorganisms. To expand their screening approach and find saline-tolerant microorganisms from different habitats, the researchers turned to insects. They discovered that the gut of certain insects, such as *Costelytra zealandica*, creates a unique alkaline microenvironment, which can be used as a source for alkaline protease separation(17, 18). During this separation process, some researchers have found that some insect intestinal endosymbionts can tolerate alkaline environment (4, 19, 20). Therefore, it was speculated that screening insect samples from special habitats could yield valuable insights into saline-tolerant microorganisms.

For this study, different species and feeding species of Coleoptera insects distributed in different salt-alkaline regions (gobi, grassland, mountain) in arid and semi-arid climate of Xinjiang were selected. These included phytophagous insects (Tenebrionidae and *Meloe*), saprophagous insects (*Scarabaeinae*), and carnivorous insects (*Blaps*, *Prosodes*, *Blatyscelis*, and *Opatrum*). The elucidation of the composition and function of endosymbiont bacteria associated with insects in arid and semi-arid climates was achieved through the utilization of microbial diversity analysis and functional prediction of the microbiota. The saline-alkali tolerance of culturable endosymbiont bacteria was verified by setting different medium conditions.

We sequenced the whole genome of a culturable endosymbitic bacterium *Brevibacterium casei* G20 (*B. casei* G20) (PRJNA754761), which can degrade cellulose and is highly tolerant to saline-alkali environment. At the same time, metabolomics and proteomics methods were used to analyze the changes of metabolites and related protein expression of the strain under normal and saline environment, and further explore the influence of the endosymbiont bacteria of Coleoptera insects on the host, providing a reference for the study of the interaction between the endosymbiont bacteria of Coleoptera insects and the host.

## MATERIALS AND METHODS

### Sample collection

Scarabaeinae, Tenebrionidae, and *Meloe* were collected in Mulei Kazak Autonomous County, Changji Hui Autonomous Prefecture, Xinjiang Uygur Autonomous Region (90.260842° E, 43.720441° N). *Blaps* and *Prosodes* were collected in Kuke Agaash Township, Fuhai County, Xinjiang Uygur Autonomous Region (87.691478° E, 47.018932° N). *Blatyscelis* and *Opatrum* were collected in Daxigou Reservoir, Urumqi County, Xinjiang Uygur Autonomous Region (87.210455° E, 43.319183° N). The same kind of insects are collected from the same area, and before the collection, the insects are in the living state of natural life. Insect samples were packaged in sterile sealed bags and partially transported to the laboratory on dry ice. Before DNA extraction, the samples were stored at −80℃. Some of them were transported to the laboratory at room temperature for screening of culturable endosymbiont bacteria. Species identification of insect samples based on apparent morphology.

### Medium

Unless otherwise specified, all strains were subjected to microbial growth physiology and phenotypic characterization studies under the same laboratory conditions in Luria-Bertani (LB) medium. LB medium was chosen due to its simple composition, easy availability, and low cost. The composition of LB medium consisted of 10 g/L tryptone, 5 g/L yeast extract, 10 g/L sodium chloride, with a pH of 7.0. Growth at varying salt concentration was determined by adding NaCl at a final concentration of 1%–10% (wt/vol) and incubated at 37℃ for 72 h. Similarly, pH range was examined in LB broth prepared in potassium phosphate buffer (50 mM) in the pH range 7.0–10.0 in steps of 1.0 pH units at 37℃ for 72 h (21). The pH values were verified after autoclaving.

### Sample treatment and endosymbiont bacteria screening

According to the method of Cooper, Ruan, Marcin et al. (22–24), the surface of the Scarabaeinae sample was washed with sterile water and soaked in 75% ethanol solution for 30 s, and the beetle body was removed, soaked in 1% sodium hypochlorite for 5 min, and rinsed with sterile water for 10 times. Then, the sterilized insects body was cut open from the anus side with sterilized anatomical scissors, the abdomen was cut open with sterilized anatomical needle, and the middle and back intestines of the insects body were taken out and ground in a sterilized mortar. Then, after grinding with aseptic grinding rod for 1 min, adding 5 mL aseptic water to make bacterial suspension, using a pipette gun to transfer bacterial suspension to 50 mL aseptic screw centrifuge tube, adding 20 mL aseptic water, oscillating at 37℃ and 150 r/min for 30 min, taking 100 μL, and coating on LB solid medium and culture at 37℃ for 24 h. After that, colonies of different shapes, colors, and sizes were selected and purified by lines. Finally, single colonies were selected with an aseptic inoculation ring and inoculated on LB liquid medium for 150 r/min for 48 h. Five hundred microliters of bacterial solution was mixed with 40% glycerol solution and placed a in frozen storage tube for storage at −80℃.

### Determination of saline-alkali tolerance of insect endosymbiont bacteria

The preserved strains were inoculated in LB liquid medium at 3% (vol/vol) and cultured at 37℃, 150 r/min for 16 h to exponential growth stage as seed liquid. The seed solution was inoculated into LB liquid medium (10 g/L NaCl, pH 7.0) and saline-alkali LB liquid medium (60 g/L NaCl, pH 10.0) at 3% (vol/vol) inoculation rate and cultured at 37℃, 150 r/min for 72 h. During this period, the culture solution $OD_{600}$ and pH value were measured every 4 h. *Brevibacterium casei* G20 seed liquid was inoculated in LB liquid medium with different saline-alkali gradients at 3% inoculation rate and cultured at 37℃ and 150 r/min for 72 h, and the culture solution $OD_{600}$ was measured.

## DNA extraction, PCR amplification

Microbial DNA was extracted from all insect samples using the E.Z.N.A.DNA kit (Omega Bio-tek, Norcross, Ga, USA). DNA concentration and purity were then determined using a NanoDrop 2000 ultraviolet-visible spectrophotometer (Thermo Scientific, Wilmington, USA). The hypervariable region V3-V4 of the bacterial 16S rRNA gene was amplified with primer pairs 338F (5′-ACTCCTACGGGAGGCAGCAG-3′) and 806R (5′-GGAC-TACHVGGGTWTCTAAT-3′) with an ABI GeneAmp 9700 PCR thermocycler (ABI, CA, USA) (25). Bacterial DNA extraction kit (Tiangen) was used to extract microbial DNA. Bacterial 16S rRNA was amplified using primers 27F (5′-AGAGTTTGATCCTGGCTCAG-3′) and 1492R (5′-TACGGTTACCTTGTTACGACTT-3′).

PCRs were performed in triplicate as follows: 4 µL of 5 × FastPfu buffer, 2 µL of 2.5 mM deoxynucleoside triphosphates, 0.8 µL of each primer (5 µM), 0.4 µL FastPfu polymerase, 0.2 µL bovine serum albumin, 10 ng template DNA, and double-distilled water (ddH$_2$O) were combined into a total volume of 20 µL. The PCR amplifications were performed as follows: 95℃ for 3 min, followed by 30 cycles of 95℃ for 30 s, 55℃ for 30 s, and 72℃ for 45 s, and a final extension of 72℃ for 10 min.

Purification of PCR products was commissioned from Shanghai Meiji Biomedical Technology Co., Ltd., and high-throughput sequencing was performed based on Illumina Miseq platform (Illumina, San Diego, USA).

## Illumina MiSeq sequencing

Purified amplicons were pooled in equimolar amounts and paired-end sequenced (2 × 300) on an Illumina MiSeq platform (Illumina, San Diego, CA, USA) according to standard protocols of Majorbio Bio-Pharm Technology Co., Ltd. (Shanghai, China).

## *Brevibacterium linens* YS sequencing sample preparation

*Brevibacterium linens* YS was inoculated in LB liquid medium at 3% (vol/vol) and cultured at 37℃ and 150 rpm/min for 16 h to exponential growth stage as seed liquid. The seed solution was inoculated in LB liquid medium and saline-alkali LB liquid medium at 3% (vol/vol) inoculum, respectively, and cultured at 37℃ and 150 r/min. In LB liquid cultures 1 h (adaptation phase), 16 h (index phase), 36 h (plateau phase), and 72 h (decline phase), an appropriate amount of bacterial solution was taken as normal environmental samples. An appropriate amount of bacterial solution was taken as saline-alkali environment samples at the 2 h (adaptation phase), 20 h (index phase), 56 h (plateau phase), and 72 h (decline phase) of saline-alkali LB liquid culture-medium. The bacterial solution was transferred to a 50-mL sterile centrifuge tube, centrifuged at 6,000 r/min at 4℃, and the supernatant was removed. Sterile PBS was washed in sterile saline (60 g/L NaCl), 6,000 r/min and centrifuged at 4℃; the supernatant was removed. The solution was frozen in liquid nitrogen for 30 min and placed at −80℃.

## *Brevibacterium linens* YS whole-genome sequencing, assembly and analysis

*Brevibacterium linens* YS DNA is extracted using the bacterial genomic DNA extraction kit (Tiangen). The DNA-TE solution is subjected to quality inspection using a spectrophotometer to ensure it meets quality standards (OD$_{260/280}$ = 1.8–2.0, OD$_{230/260}$ = 1.6–1.7). Qualified DNA-TE solution is sent to BGI Genomics Co., Ltd. in Shenzhen for sequencing on the Illumina HiSeq 4000 platform. The raw reads underwent quality control processing, involving the removal of low-quality bases and the elimination of any adapters from clean reads. This was accomplished using Trimmomatic v0.36 (26). Afterward, the reads that passed the quality control criteria were used for *de novo* assembly using the Unicycler v0.4.8 assembler, which is specifically designed for Illumina short-read data. The assembly quality was evaluated using the Quality Assessment Tool for Genome Assemblies (QUAST) v5.0.2. To annotate the genome, the Rapid Annotation Subsystems Technology (RAST) toolkit was employed, which utilizes a k-mer-based

search method against CoreSEED. To visually represent the genome, a circular genome map was generated using CGView.

## *Brevibacterium linens* YS metabolite and total protein extraction

Bacterial metabolite extraction, LC-MS detection, and compound identification were entrusted to Shenzhen Bada Gene Technology Co., LTD. TMT proteome sequencing was commissioned by Beijing Novogene Biotechnology Co., LTD. After thawing slowly at 4℃, take 25 mg and weigh it to 1.5 mL Ebendorf test tube, add 800 µL extract (methanol:acetonitrile:water = 2:2:1, vol:vol:vol, −20℃ pre-cooling) + 10 µL internal standard, add two small steel balls, and grind in a tissue grinder at 50 Hz for 5 min. Ultrasonicate in a water bath at 4℃ for 10 min and refrigerate at −20℃ for 1 h. Centrifuge at 25,000 rpm for 15 min at 4℃. After centrifugation, 600 µL of supernatant was taken, dry in a frozen vacuum concentrator, and 200 µL of complex solution (methanol: water = 1:9, vol:vol) was added for re-dissolution, vortex vigorously for 1 min, ultrasonic bath at 4℃ for 10 min, centrifugation at 25,000 rpm at 4℃ for 15 min. The supernatant was placed in a sample bottle. Twenty microliters of supernatant of each sample was mixed into QC samples to evaluate the repeatability and stability of LC-MS analysis process.

## Liquid chromatography and mass spectrometry parameters

The chromatography was performed on BEH C18 column (1.7 µm, 2.1*100 mm, Waters, USA). In the positive ion mode, the mobile phase consisted of aqueous solution containing 0.1% formic acid (liquid A) and 100% methanol containing 0.1% formic acid (liquid B). In the negative ion mode, the mobile phase consisted of aqueous solution containing 10 mM formic acid (liquid A) and 95% methanol containing 10 mM formic acid (liquid B). The following gradient was used for elution: 0–1 min, 2% B solution; 1–9 min, 2%–98% B solution; 9–12 min, 98% B solution; 12–12.1 min, 98% B liquid ~2% B liquid; 12.1–15 min, 2% B solution. The flow rate was 0.35 mL/min, the column temperature was 45℃, and the sample size was 5 µL.

Q Exactive HF mass spectrometer (Thermo Fisher Scientific, USA) was used for primary and secondary mass spectrum data acquisition. The mass/nucleus ratio was 70–1,050, the primary resolution was 120,000, the AGC was 3e6, and the maximum Injection time was 100 ms. According to the parent ion strength, Top3 was selected for cracking, and the second-level information was collected. The second-level resolution was set to 30,000, the AGC was set to 1e5, the maximum injection time was set to 50 ms, and the stepped nce was set to 20, 40, and 60 eV. Ion source (ESI) parameter Settings: Sheath gas velocity (Sheath gas flow rate) of 40, auxiliary gas flow velocity (Aux gas flow rate) for 10, Spray electric [Spray voltage (∣∣ KV)] positive ion mode is 3.80, and negative ion mode is 3.20. The ion transfer tube temperature (Capillary temp) is 320℃, and the auxiliary gas heater temp is 350℃.

## Data analysis

We used Uparse (version 7.1) to analyze the data of the 16S rRNA gene. The obtained sequences were denoised, unified, and aligned into OTUs using definitions of ≥97% sequence similarity. The taxonomy of each OTU representative sequence was analyzed by RDP (Ribosomal Database Project) Classifier (v.2.11) against the 16S rRNA database (Silva SSU138) using the confidence threshold of 0.7.

Mothur (v.1.30.2) performed Alpha diversity analysis to reflect the species richness and diversity within the sample. Qiime (v.1.9.1) performed Beta diversity analysis to compare the community composition of the tested samples. The results of 16s rRNA sequence were BLAST compared, and MEGA X software was used to construct the phylogenetic tree by neighbor-joining method (NJ) cluster analysis. Compound Discoverer 3.0 (Thermo Fisher Scientific, USA) software was used for LC-MS data processing, mainly for peak extraction, peak alignment, and compound identification. Compound identification database has a BGI Library (https://mscloud.bgi.com), mzCloud

([https://www.mzcloud.org](https://www.mzcloud.org)), chemSpider ([http://www.chemspider.com](http://www.chemspider.com)). Metabolomics R software package and ChiPlot ([https://www.chiplot.online](https://www.chiplot.online)), metabolome information analysis process were used for data preprocessing, statistical analysis, metabolite classification annotation, and functional annotation. Proteomics is based on the Raw files detected by mass spectrometry, searching the corresponding database, identifying the protein from the database search results, and analyzing the mass tolerance distribution of peptide, protein, and parent ion to evaluate the quality of the mass spectrometry detection data. The common functional databases of the identified proteins were annotated, including COG database, GO database, KEGG database, etc. Then, the quantitative analysis of proteins was carried out, including the overall difference analysis of identified proteins, the screening of differential proteins, and the clustering analysis of expression patterns. Finally, a series of functional analyses of differential proteins, such as interaction network analysis, were performed on the screened differential proteins.

## RESULTS

### Identification of insect morphology and species

Under the guidance of Professor Hu Hongying, an expert in entomology, the identification of insect specimens was carried out (Fig. 1) based on their external morphological characteristics, such as body length, body color, antenna morphology, and pronotum features. Various reference materials, including the "Colored Pictorial Handbook of Insects in XinJiang" (27), were utilized during the identification process. It was determined that all of these insect specimens belong to the order Coleoptera. One species of insect was identified at the family level as Tenebrionidae (28), while another species was identified as Scarabaeinae (29). The remaining five insect specimens were identified at the genus level, namely, *Meloe* (30), *Blaps*, *Prosodes* (31), *Blatyscelis*, and *Opatrum* (32).

### Isolation and identification of culturable endosymbiont bacteria from Coleoptera

The culturable endosymbiont bacteria selected from the samples of Coleoptera were isolated and cultured. Forty-eight strains of bacteria were isolated by observing the

| Feeding habit | Putrefaction | Herbivory | | Predatory behavior | | | |
|---|---|---|---|---|---|---|---|
| Identification level | Scarabaeinae | Tenebrionidae | *Meloe* | *Blaps* | *Prosodes* | *Blatyscelis* | *Opatrum* |
| Front | | | | | | | |
| Back | | | | | | | |

FIG 1   Identification results of insect samples.

colony characteristics and cell morphology. 16S rRNA sequences were sequenced for these bacteria, and the 16S rRNA sequences of 48 strains of bacteria were constructed by NJ cluster analysis using MEGA-X software (Fig. 2). The results showed that 48 strains of bacteria belonged to 3 phyla in the biological classification, namely, *Firmicutes*, *Proteobacteria,* and *Actinobacteria*. *Firmicutes* and *Proteobacteria* had a high abundance of 24 and 20 strains, respectively, while the smallest *Actinobacteria* was only 4 strains. The strains of *Pseudomonas* and *Bacillus* were isolated and screened in the samples of genus level classification. In addition to the above genera, a more widely distributed genus *Pantoea* was isolated from all the samples except the *Meloe*.

## Determination of saline-alkali resistance of culturable endosymbiont bacteria in Coleoptera

In order to study the salt-alkali resistance of culturable endosymbiont bacteria of Coleoptera insects, the salt-alkali tolerance of 48 strains of insect endosymbiont bacteria was determined. Figure 3 is a heat map constructed according to $OD_{600}$, which was grown for 72 h under different salt concentrations and pH values, to evaluate the strain's tolerance to saline-alkali environment. It can be seen that 48 strains of cultivable endosymbiont bacteria have a certain tolerance to saline-alkali environment and can prolifate in pH = 7–8 and 1%–3% NaCl environment. In Fig. 3A, the salt-alkali tolerance of cultivable endosymbiont bacteria was correlated with samples of Coleoptera. It was found that individuals with prominent salt-alkali tolerance were present in different samples of Coleoptera insects. Among them, individuals of the *Meloe* showed slightly

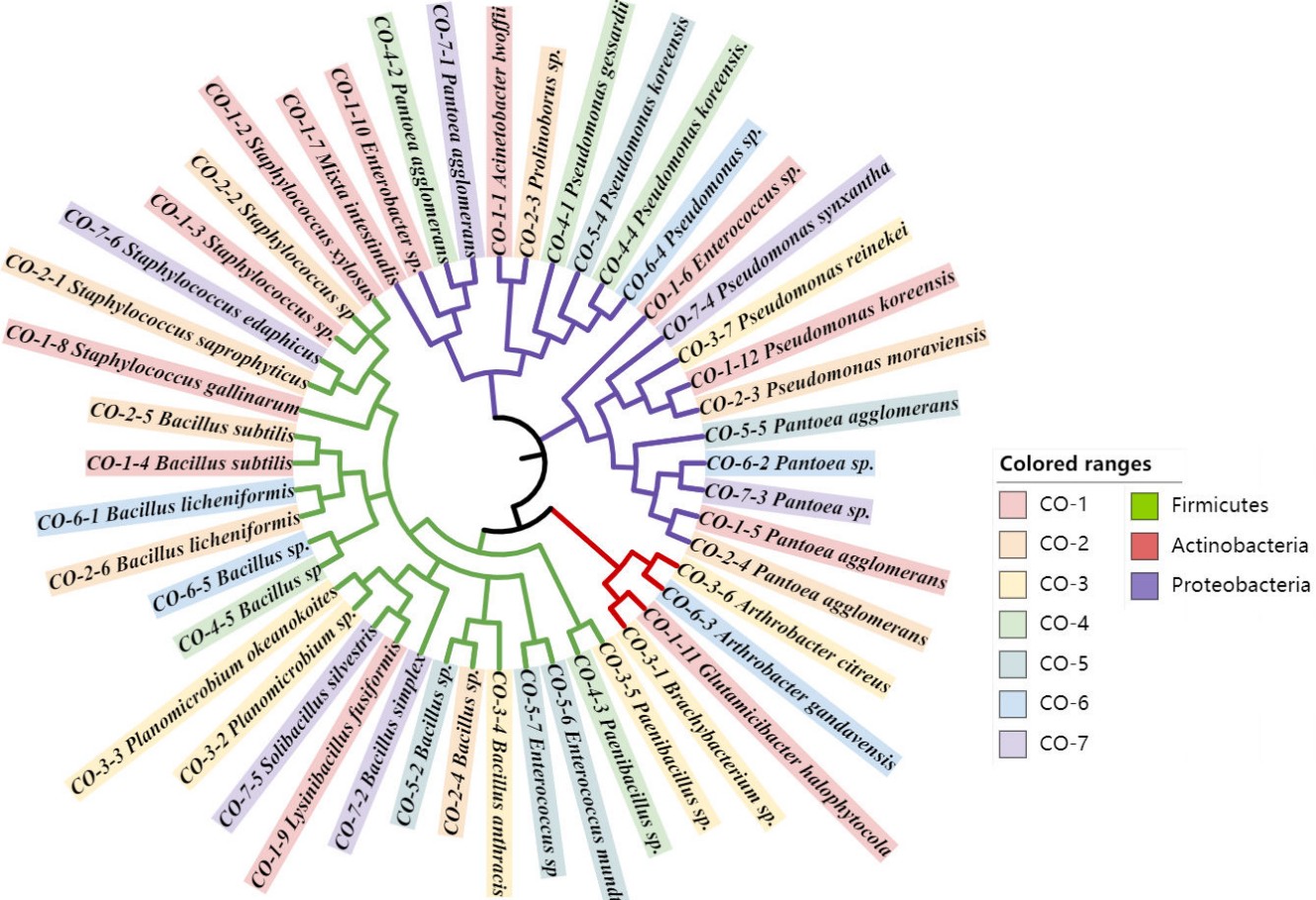

**FIG 2** Phylogenetic relationships among culturable endosymbiont bacteria in Coleoptera insect samples stands based on the 16S rDNA sequences. CO-1：Tenebrionidae; CO-2：Scarabaeinae; CO-3：*Meloe*; CO-4：*Blaps*; CO-5：*Prosodes*; CO-6：*Blatyscells*; CO-7：*Opatrum*.

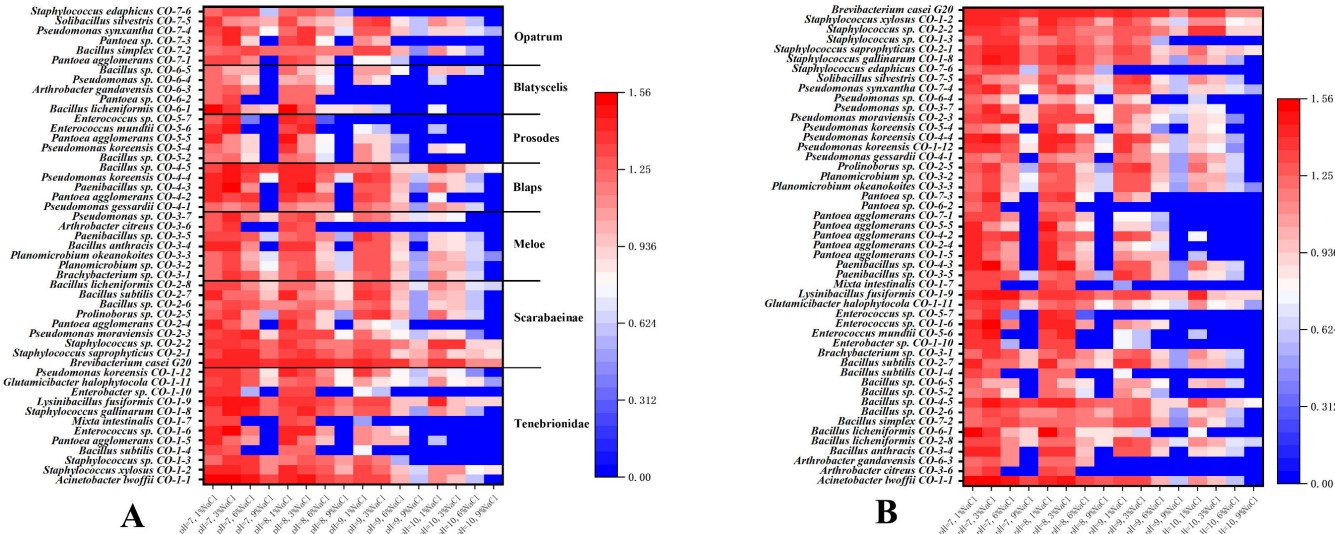

**FIG 3** Heat map of saline alkali tolerance of culturable endosymbiont bacteria in Coleoptera insect samples. (A) Association analysis of endosymbiont bacteria salt-alkali tolerance and Coleoptera insect samples that can be cultivated. (B) Association analysis of endosymbiont bacteria salt-alkali tolerance and strain species that can be cultivated.

weaker salt-alkali tolerance, while individuals of the *Prosodes* and *Blatyscelis* exhibited the weakest salt-alkali tolerance among the samples. In these two samples, cultivable endosymbiont bacteria could hardly grow in environments with pH 10% or 9% NaCl. On the other hand, individuals of the Scarabaeidae insects had the highest overall salt-alkali tolerance in terms of the screened cultivable endosymbiont bacteria, and this may be closely related to the habitat selection and food preferences of the Scarabaeinae (33). Figure 3B correlates the salt-alkali tolerance of cultivable endosymbiont bacteria with their species and genera. It was found that *Pseudomonas* and *Staphylococcus* exhibited the highest tolerance to salt-alkali environments and could grow normally in environments with pH 7–10 and 1%–6% NaCl. Some strains, such as CO-7–4 and CO-1–2, maintained their tolerance even in environments with pH 10% and 9% NaCl. In contrast, *Bacillus* generally had slightly weaker salt-alkali tolerance although certain strains like CO-4–5 exhibited strong tolerance and could grow in environments with pH 10% and 9% NaCl. *Pantoea*, on the other hand, were unable to grow in environments with pH 10% or 9% NaCl. Overall, when considering the growth performance of all strains in salt-alkali environments, it can be observed that most strains can grow in low-salt environments when the pH is 10, indicating that their tolerance to alkaline conditions is higher than their tolerance to saline conditions.

## Alpha diversity of endosymbiont bacteria communities in Coleopteran insects

In order to further understand the community composition of Coleoptera ensymbiont bacteria, we analyzed the diversity of Coleoptera insect samples by second-generation sequencing.

Alpha Diversity analysis can understand the diversity and richness of the endosymbiont bacteria community in the sample. Shannon index and Simpson index reflect microbial community diversity. The larger Shannon index is, the higher community diversity is, and the larger Simpson index is, the lower community diversity is. ACE index and Chao index reflect microbial community richness, and the larger the value, the higher the community richness.

As can be seen from Table 1, the coverage of all samples was greater than 99%, indicating high integrity of sequencing data. The Shannon index ranges from 1.4025 to 3.9706, and the ACE index ranges from 171.0398 to 982.5873. The Shannon index and

**TABLE 1** Alpha diversity index of endosymbiont bacteria of Coleoptera insect samples

| Species classification | OTU | Coverage | Alpha diversity | | | |
|---|---|---|---|---|---|---|
| | | | Shannon | Simpson | ACE | Chao |
| Tenebrionidae | 358 | 0.9969 | 3.5862 | 0.1008 | 565.0752 | 566.4166 |
| Scarabaeinae | 271 | 0.9992 | 2.0962 | 0.1867 | 171.0398 | 158.1111 |
| *Meloe* | 466 | 0.9995 | 2.4219 | 0.3088 | 481.1776 | 483 |
| *Blaps* | 257 | 0.9978 | 1.4025 | 0.4841 | 379.7980 | 401.2857 |
| *Prosodes* | 206 | 0.9985 | 1.5466 | 0.3561 | 432.3590 | 345.5333 |
| *Blatyscelis* | 727 | 0.9969 | 3.9706 | 0.0776 | 982.5973 | 979.0088 |
| *Opatrum* | 389 | 0.9984 | 2.4295 | 0.2707 | 517.4537 | 522.1132 |

ACE index of *Blatyscelis* insect samples were the highest, indicating that the microbial community diversity and richness of *Blatyscelis* insect samples were the highest. The Shannon index of *Blaps* and *Prosodes* samples was low, indicating low microbial community diversity. The ACE index of Scarabaeinae was the lowest, indicating that the community richness of Scarabaeinae was low. The results of culturable endosymbiont screening showed that the endosymbionts of Scarabaeinae generally exhibited higher saline-alkali tolerance. This was attributed to the high saline-alkaline nature of the endogenic environment in the samples, leading to a correspondingly limited microbial diversity (34).

## Composition of endosymbiont bacteria community of Coleoptera insects

Using Uparse (v.7.1) at 97% similarity, the endosymbiont bacteria of Coleoptera samples were coclustered to 1416 OTUs, which were identified as 24 phyla, 66 classes, 164 orders, 278 families, 548 genera. Figure 4A shows the community structure of each Coleoptera sample at phylum level, with those with less than 1% abundance combined as "others." Phyla with an abundance of more than 1% include *Proteobacteria*, *Firmicutes*, *Actinobacteria*, *Cyanobacteria Chloroflexi*, and *Deinoc occus-Thermus*, *Patescibacteria*. *Proteobacteria* occupies a high abundance proportion in all samples, ranging from 43.15%—94.6%. In addition, *Actinobacteria* and *Firmicutes* also had a certain abundance ratio in all samples, ranging from 2.99%—33.9% and 1.3%—50.9%. They were the dominant bacteria in the samples of the Tenebrionidae, the *Blatyscelis,* and the Scarabaeinae, respectively. In order to understand the overall composition of the endosymbiont community of Coleoptera insects, the average population abundance of each insect sample phylum was taken in Fig. 4B to compare the community composition. It can be found that *Proteobacteria*, *Firmicutes,* and *Actinobacteria* are absolutely dominant in the endosymbiont bacteria composition of Coleoptera insects, accounting for 66.13%, 17.07%, and 14.33% of the total abundance, respectively. The sum of the three abundances was 97.53%, which was highly similar to the classification results of culturable endosymbiont bacteria screened from insect samples.

Figure 5A observed the composition similarity and overlap of OTUs of all insect samples using Venn chart and found that there were 79 intersecting OTUs in 7 groups of insect samples. Each sample had unique OTUs, of which 340 OTUs were found in *Blatyscells* samples, which was related to the high microbial community diversity and community richness in tibialis samples. Figure 5B shows the proportion of annotations at the genus level for 79 intersecting OTUs. It was found that the proportion of *Pseudomonas* was the highest (44.17%), followed by *Solibacillus* (10.63%) and unclassified genus under *Enterobacterales* (8.57%). In the screening of culturable endosymbiont bacteria, the strain of *Pseudomonas* can be isolated and screened from each Coleoptera insect sample, which further indicates that the strain belonging to the *Pseudomonas* has a high abundance proportion in the endosymbiont bacteria of Coleoptera insects in semi-arid areas. *Bacillus* accounted for a relatively low proportion of 2.06% compared with *Pseudomonas*. However, *Bacillus* was also screened from samples of various Coleoptera insects. Combined with the determination of saline-alkali tolerance of culpable

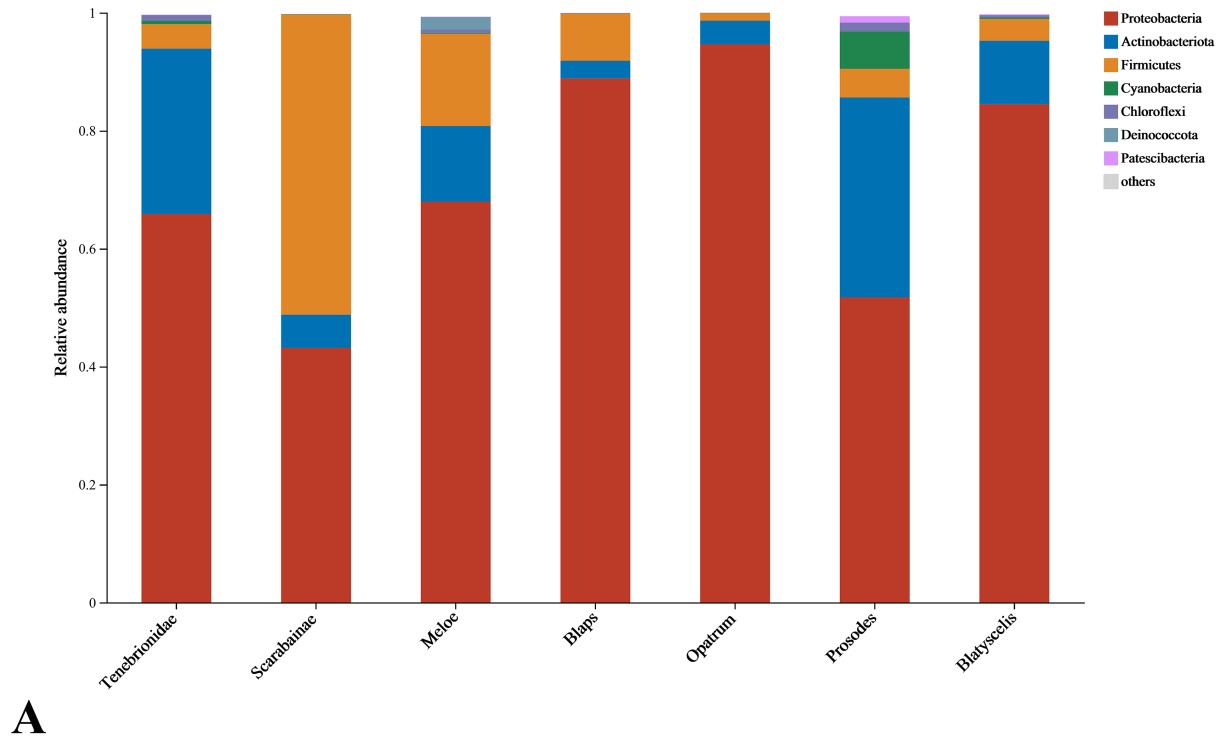

**A**

Community analysis pieplot on Phylum level :Coleoptera

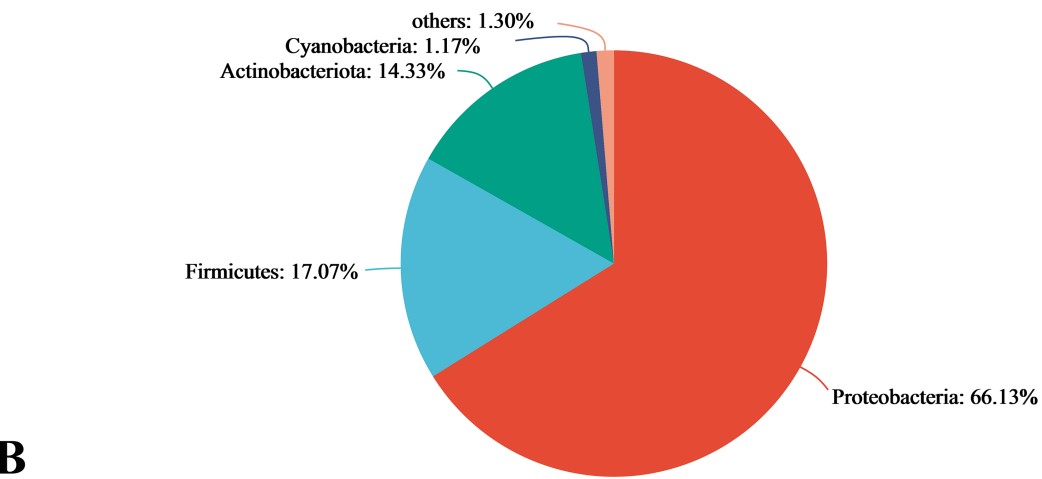

**B**

**FIG 4** Community structure of endosymbiont bacteria in each of Coleoptera insect samples at phylum level (A). Community structure of endosymbiont bacteria in Coleoptera insect samples at phylum level (B).

endosymbiont bacteria, the *Pseudomonas* strain was not only the culpable bacteria of all the insect samples but also the one with the strongest saline-alkali tolerance. *Bacillus* strain is similar to *Pseudomonas* strain, but its saline-alkali tolerance is slightly weak. The difference is that *Bacillus* strain was screened more in the endosymbiont bacteria of Scarabaeinae samples, while *Pseudomonas* was distributed evenly.

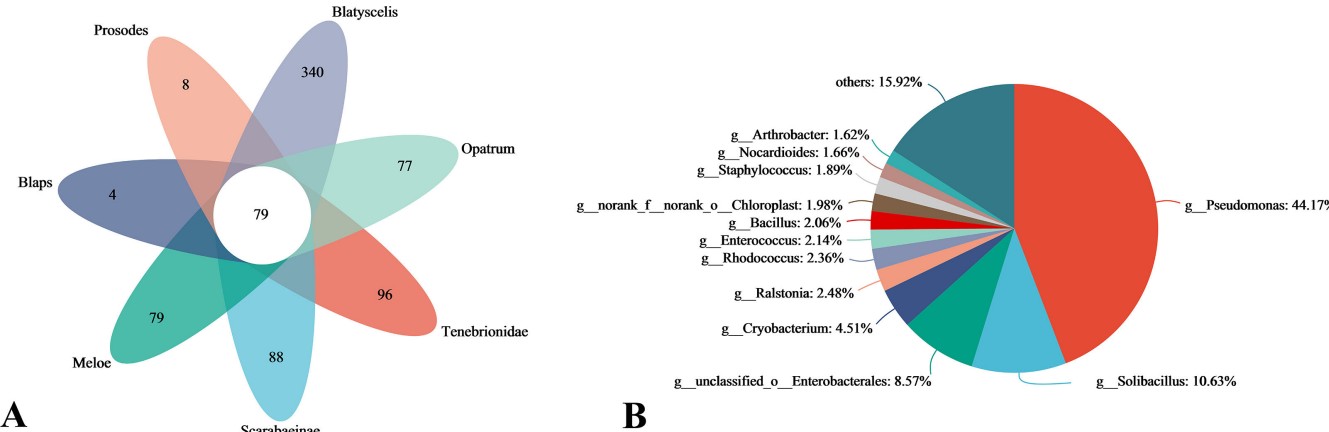

**FIG 5** Venn diagram of OTU under in endosymbiont bacteria of Coleoptera insect samples (A). Proportion of endosymbiont bacteria in Coleoptera insect samples at genus level (B).

## Function prediction and analysis of endosymbiotic bacteria and culturable endosymbiont bacteria in Coleoptera

The endosymbiont community structure information of different insect samples obtained from the annotation of sequencing sequence analysis was compared with PICRUSt software to predict the abundance of functional genes contained in each sample, so as to obtain information on how the metabolic function of endosymbiont affects the host. Figure 6A shows that the COG of each Coleoptera sample has higher function in amino acid transport and metabolism, energy production and conversion, and carbohydrate transport and metabolic pathway abundance. The results of amino acid transport and metabolism fully complement the differential enrichment pathways of *B.casei* G20 metabolomics and proteomics in saline environment. In addition, the transport and metabolism of carbohydrates are closely related to the diet of lower Coleoptera insects in arid and semi-arid areas, where Coleoptera insects mostly live on cellulose-rich wastes such as straw and herbivorous animal feces, which also makes it an effective way to mine efficient cellulase from insect intestinal environment.

Scarabaeinae samples showed the best performance in the determination of salinization capacity of cultivable endosymbiont bacteria, so as a typical sample, functional annotation of this sample, was analyzed with emphasis through COG database (Fig. 6B). In the COG database notes, except amino acid transport and metabolism, the inorganic ions transport and metabolism (P) also noted a relatively high abundance, indicating that the inorganic ions can reduce the high concentration of ions accumulated in the saline environment and maintain the osmotic pressure balance by enhancing the inorganic ions transport and metabolism pathway. This is the expression of endosymbiont bacteria salt-alkali tolerance in molecular transport function.

## Salt-alkali tolerance of *Brevibacterium casei G20*

A strain of *Brevibacterium* with strong saline and alkali tolerance was screened from the LB selective medium with high salinity and alkalinity which can survive under the extreme conditions of 11% NaCl and pH 10. We also obtained a genome completion map of this strain(Fig. 7A), named it G20 (GenBank accession no. PRJNA754761). In order to test its salt and alkali tolerance, we carried out liquid shake flask fermentation and solid plate spotting. Figure 7B is the growth curve test of *Brevibacterium casei* G20 in normal LB liquid medium (pH 7, 10 g/L NaCl) and high-salinity and alkinity LB liquid medium (pH 10, 60 g/L NaCl). It can be seen that under high pH and in a high-salt environment, *Brevibacterium casei* G20 took more time to adapt to the saline-alkali environment during the adaptation period, and when it reached the plateau period, the growth density of the bacteria was only slightly lower than that in the normal environment. Figure 7C shows

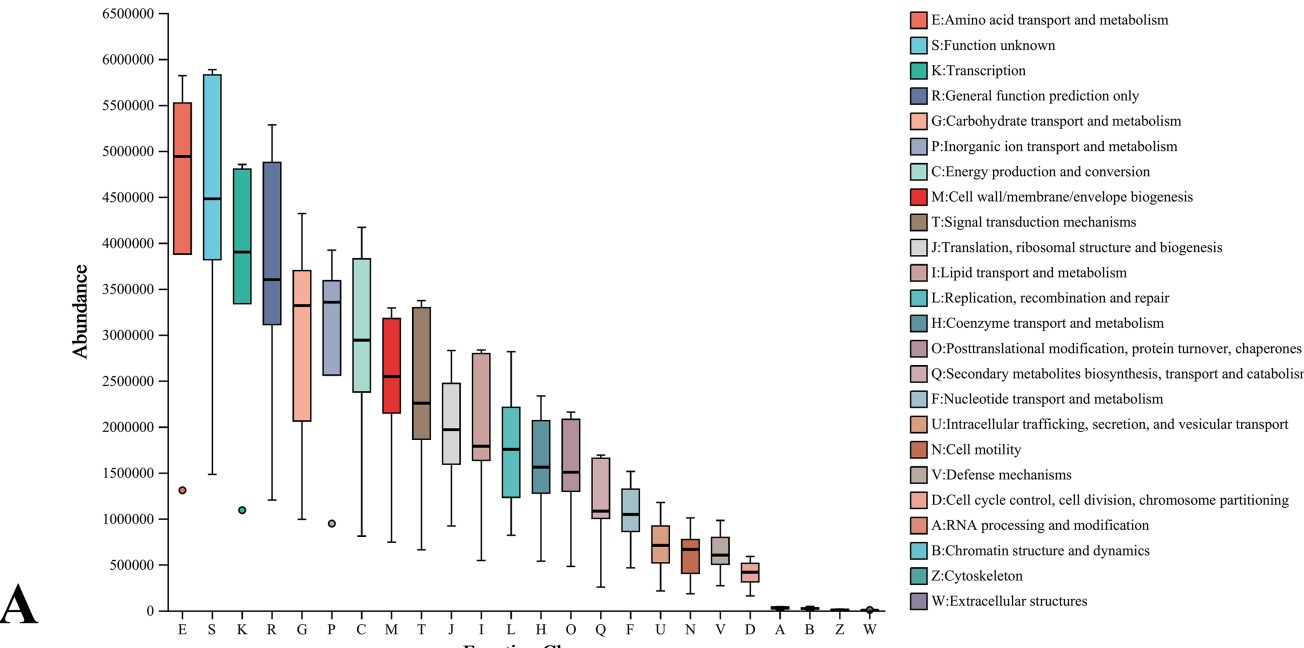

COG function classification : Scarabaeinae

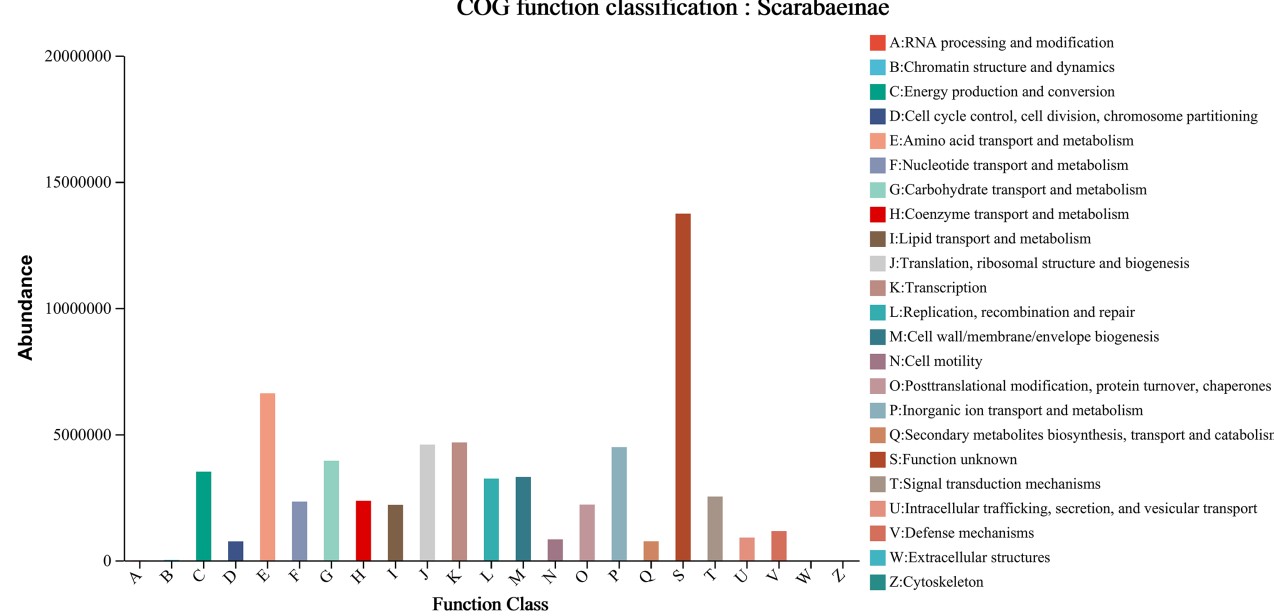

**FIG 6** Annotated box diagram of the COG database of endosymbiont bacteria in Coleoptera insect samples (A). Annotations to the COG database of endosymbiont bacteria in Scarabaeinae samples (B).

the colony growth on LB solid plates at different salt concentrations and pH. It can be seen that under normal circumstances, the color of the colony is milky white, the shape is round, and the surface is smooth. The colony can still grow under the condition of 60 g/L NaCl and pH 10. Since these results show that *B. casei* G20 has a high salinity tolerance, we can try to use it as a chassis strain for research and industrial production.

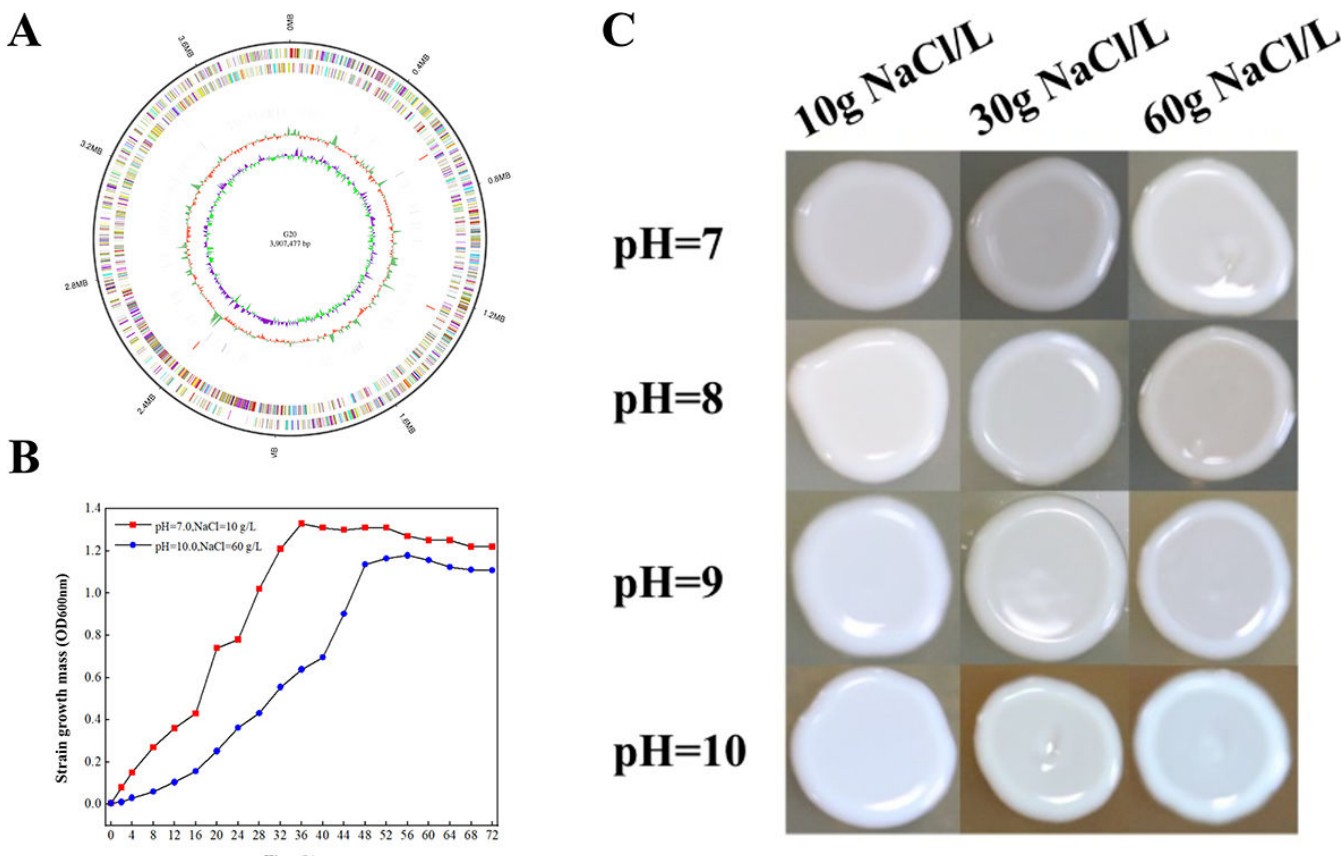

**FIG 7** Circular genome view of *Brevibacterium casei* G20 (A) and salt-alkali tolerance of *Brevibacterium casei* G20. (B) Growth curve of *Brevibacterium casei* G20 under different salinity and alkalinity conditions. (C) Colony growth of *Brevibacterium casei* G20 under different salinity and alkalinity levels.

## Metabolomics analysis of the Coleoptera endosymbiont bacterium *B. casei* G20

In order to further explore the mechanism of the influence of endosymbiont bacteria in Scarabaeinae samples influence the host under saline-alkali environments in arid and semi-arid areas, the metabolites of a highly saline-alkali tolerant individual, *B. casei* G20, were isolated and identified during four growth stages in normal and saline-alkali environments. Table 2 presents the quantities of metabolites at different growth stages under normal and saline-alkali environments. By combining positive and negative ion models, it was observed that the number of up-regulated and down-regulated metabolites during the exponential and plateau stages was greater compared to other

**TABLE 2** Different metabolites in different growth stages under normal and saline-alkali environments in *B.casei* G20[a]

| Sample group | Mode | Number of differential metabolites | Up | Down |
| --- | --- | --- | --- | --- |
| SA-lag : CG-lag | pos | 1,885 | 891 | 994 |
| | neg | 771 | 398 | 373 |
| SA-log : CG-log | pos | 2,736 | 1,119 | 1,617 |
| | neg | 983 | 499 | 484 |
| SA-sta : CG-sta | pos | 3,343 | 1,234 | 2,109 |
| | neg | 999 | 429 | 570 |
| SA-dec : CG-dec | pos | 2,039 | 980 | 1,059 |
| | neg | 1,067 | 403 | 664 |

[a]Fold change ≥ 1.2 was significantly up-regulated, and fold change ≤ 0.83 was significantly down-regulated; $q < 0.05$.

growth cycles, with the plateau stage showing significant advantages. Therefore, only the differential metabolite lineages in the plateau stage were discussed. Under saline-alkali stress, 2,736 and 983 metabolites with significant differences were detected using positive and negative ion modes, respectively, during the exponential stage. At the plateau stage, 3,343 and 999 metabolites with significant differences were detected in positive and negative ion modes, respectively. These differential metabolites mainly include organic acids and their derivatives, such as indole-3-acetic acid; phenyl ring compounds, such as tetramethyl-*p*-phenylenediamine; organic heterocyclic compounds, etc. Bar charts were created to display pathways with significant enrichment of differentiated metabolites during the plateau period, as shown in Fig. 8. In the plateau phase, the differential metabolites are mainly involved in the metabolic pathway, secondary metabolites, and antibiotic biosynthesis pathways, as indicated by the positive and negative ion models. Amino acid biosynthesis pathway is the common differential metabolic pathway in the positive and negative ion models during this period, and this pathway has outstanding performance in the number of differential metabolites, enrichment factors, and difference significance. Previous studies have demonstrated that amino acid metabolic pathways play a crucial role in response to salt stress and the tolerance of plants to ammonium (35, 36). It can be inferred that amino acid biosynthesis pathway is a very important metabolic pathway in response to saline environment of *B. casei* G20.

## Proteomic analysis of Coleoptera endosymbiont bacterium *B. casei* G20 strain (based on differential metabolic pathway)

To more effectively elucidate the differential expression of strain-specific proteins, the up-regulated differential proteins were selectively identified based on a fold change (FC) threshold exceeding 1.2 and a significance level (*P*-value) below 0.05, whereas the down-regulated differential proteins were identified based on a fold change below 0.83 and a significance level (*P*-value) below 0.05. Consequently, the analysis revealed 178 up-regulated differential proteins and 299 down-regulated differential proteins.

Under saline-alkali stress, the expression of *B. casei* G20 strain was observed to upregulate several microbial metabolism pathways, including Propanoate metabolism and Microbial metabolism in diverse environments, which were notably enriched in proteins. Furthermore, fatty acid metabolism, riboflavin metabolism, benzoate degradation, sulfur metabolism, caprolactam degradation, and limonene and pinene degradation pathways were also upregulated. Conversely, down-regulated proteins were found to be significantly enriched in RNA degradation and RNA polymerase pathways.

Based on differential metabolic pathway analysis, the effects of saline-alkali stress on amino acid metabolism and carbohydrate transport were studied (Fig. 9). According

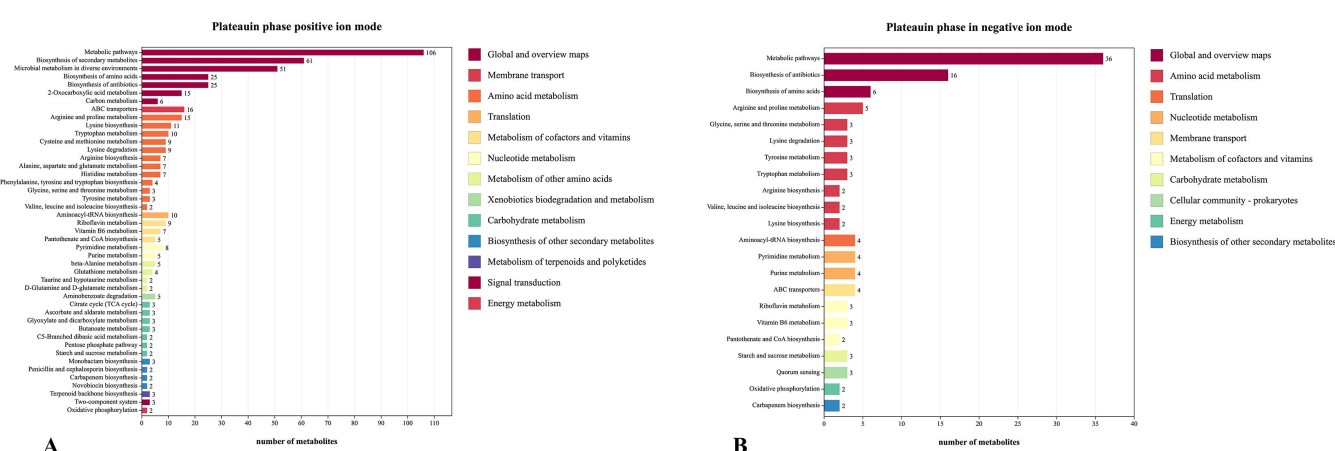

**FIG 8** Bar chart of different metabolic pathways during the plateau phase between normal environment and saline-alkali environment of *B.casei* G20 (*P* < 0.05). Plateau phase period under the positive ion model (A). Plateau phase in negative ion mode (B).

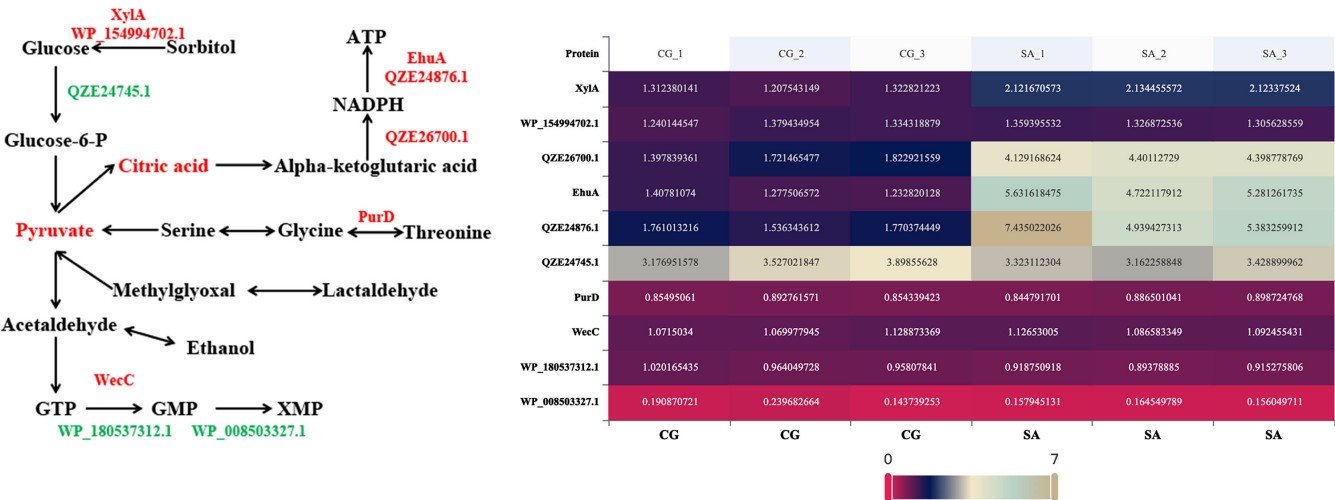

**FIG 9** Effects of *B.casei* G20 saline-alkali resistance on carbohydrate transport and protein expression related to amino acid metabolism (up-regulated differential proteins were screened with FC >0.2 and *P* < 0.05, and down-regulated differential proteins were screened with FC < −0.2 and *P* < 0.05). Protein expression is shown as up-regulated (red) or down-regulated (green).

to Fig. 9, 10 proteins are differentially expressed in these metabolic pathways, of which 7 proteins are up-regulated and 3 proteins are down-regulated. The heat map illustrates the expression trends of these differentially expressed proteins. XylA (WP_182069873.1，EC 5.3.1.5）belongs to the GH11 family and has strong alkali resistance (37). They could stable over a broad pH range (pH 6.0–11.0 and 7.0–10.0). PurD (WP_180536073.1，EC 6.3.4.13）plays a crucial role in the second step of the purine *de novo* synthesis pathway (38), and purine nucleotide metabolism is involved in energy metabolism in cell life (39). EhuA (QZE24878.1, EC 7.4.2.1) is a key energy transporter involved in the synthesis of ectoine and its derivatives, and ectoine is an important compatible substance for cell resistance to saline-alkali stress (40). Other differentially expressed proteins were involved in glycolytic metabolism and partial amino acid metabolism (glycine, serine, and threonine) of *B. casei* G20, and the riboflavin metabolism level of *B. casei* G20 was decreased.

## DISCUSSION

This study investigated the composition and function of endosymbiont bacteria in Coleoptera, the salt-alkali tolerance of culturable endosymbiont bacteria, as well as the metabolites and expressed proteins of an individual strain (*B. casei* G20).

Seven Coleoptera insects with different species and diets were selected for culturable bacteria screening. A total of 48 strains of bacteria were screened. These strains belonged to four phyla and had similar composition and structure to the endosymbiont bacteria community of Coleoptera, which proved the reliability of the screening results of culturable endosymbiont bacteria. At the genus level, among the 79 bacteria annotated by OTUs in all the samples, only *Bacillus*, with the highest proportion of 44.17%, and *Bacillus*, with the lowest proportion of 2.06%, were screened out from the culturable endosymbiont bacteria in each insect sample. In the test of saline-alkali tolerance, *Bacillus* showed the best performance of *Pseudomonas* and *Bacillus* closely behind. In the past few decades, the development of the strains of *Pseudomonas* and *Bacillus* has gradually matured, and many of its strains have developed into multifunctional microbial cell factories, which have been used for the recombinant synthesis of various high-value natural products and industrial enzyme preparations (41, 42). The resistance of *Bacillus* and *Pseudomonas* strains of Coleoptera culturable endosymbiont bacteria to saline-alkali environment makes them potentially the chassis strains of the next generation of industrial biotechnology (NGIB) (43). *Staphylococcus* and *Pseudomonas* strains showed

similar salt-alkali tolerance and accounted for a higher proportion among the cultivable bacteria. However, at the genus level, *Staphylococcus* shared by all samples noted by OTU accounted for a relatively low proportion, only 1.89%, and the endosymbiont bacteria could not be screened out in all insect samples, which was different from the result of *Bacillus*. It was speculated that the LB medium used was more suitable for the growth of *Bacillus* strains. In addition, some strains of Staphylococcus are relatively common pathogenic bacteria, and researches on them mainly focus on the pathogenic mechanism of the strains (44, 45), with few industrial applications.

In terms of the source of culturable endosymbiont bacteria, the salt-alkali tolerance of culturable endosymbiont bacteria in Scarabaeinae samples was the highest, which was speculated to be related to its feeding habits (46–48). At the same time, we found that the tolerance of culturable endosymbiont bacteria to alkaline environment was higher than that to salt environment, which may be related to the alkaline environment of insect gut, and also makes them more adaptable to the production of NGIB. In the production of NGIB, the high-salt and high-alkali environment of fermentation liquid will be used to resist the contamination of hybrid bacteria (43), and the maintenance cost of alkaline environment in fermentation is significantly lower than that of high-salt environment which needs to be maintained by massive addition of sodium chloride. Meanwhile, high concentration of salt ions will increase the separation cost of later products. Alkaline environment had little effect on the product separation.

In the functional analysis of endosymbiotic bacteria in typical Scarabaeinae samples, inorganic ions transport and metabolism, which can relieve the high concentration of ions accumulated in strains in saline-alkali environment and maintain the osmotic pressure balance of strains (49). This is the expression of endosymbiont bacteria's salt-alkali tolerance on molecular transport function. *B.casei* G20 is the best performer among the cultured endosymbiont bacteria samples from Scarabaeinae. It can grow normally in pH 10, 9% NaCl environment. At the same time, it can maintain the life process on a solid plate with microcrystalline cellulose as the only carbon source. In the analysis of metabolites, it was found that the metabolites with high content in the four growth stages of *B. Casei* were mainly sugars, amino acids and their derivatives, organic acids and their derivatives, and fatty acyl compounds. Among them, betaine has a very high content in the four periods, which can reduce the stress of drought and low-temperature environment on insects (50), but betaine, as a typical saline-tolerant microbes osmotic pressure protection agent, has no significant difference in its peak spectrum intensity between the *B.casei* G20 saline environment and normal environment. These results indicated that its synthesis was not obviously related to the regulation mechanism of osmotic pressure in *B.casei* G20 saline-alkali environment. Therefore, it is speculated that a large amount of synthetic betaine may be excreted into the extracellular system to enhance the resistance of the host. In addition, studies have shown that the contents of some sugars, amino acids, organic acids, and fatty acids in the body of phytophagous insects will decrease after they eat host plants under saline-alkali stress, and the corresponding predators on the food chain will also change (51). In addition, high contents of amino acids, organic acids, and fatty acids in the metabolites of *B. casei* G20 may be excreted into the extracellular layer to supplement the host with these compounds. In addition, *B. casei* G20 a sucrose of polyols (Bis(methylbenzylidene)sorbitol), Bis(4-ethylbenzylidene)sorbitol, free amino acids (L-glutamine, L-isoleucine), and sugars can also be obtained from G20 metabolites and can help the host regulate osmotic pressure balance and cope with low temperature environment (52).

The accumulation of saline-tolerant microorganisms in Coleoptera insects in arid and semi-arid climates suggests that Coleoptera in this environment may be an important way to screen saline-tolerant microorganisms in the future. As an important branch of extremophile, saline-tolerant microorganisms occupy an important position in industrial production. It can not only tolerate high salt environment but also survive at high pH value. It has higher robustness and environmental adaptability than ordinary

microorganisms and can better survive in the dynamic environment of constant changes (53, 54). Current studies on saline-alkali-resistant microorganisms mainly include the study on the mechanism of saline-alkali resistance of upstream saline-alkali-resistant microorganisms (55, 56), the transformation of the industrial direction of saline-alkali-resistant microorganisms (57, 58), the industrial production of downstream saline-alkali-resistant microorganisms (59), and the utilization of various saline-alkali-resistant enzymes and proteins (60, 61). In recent years, with the rise of next-generation industrial biotechnology (NGIBs), extremophiles are used as chassis cells to establish an open, non-sterilizing continuous fermentation production system to produce chemicals at low cost and high quality (62), which has gradually replaced the traditional production of some chemicals. The long-term symbiosis between cultured endosymbiont bacteria and the host also makes it a way to discover some new bioactive molecules (63–65). Meanwhile, some bioactive molecules (betaine, *cis*-4-hydroxy-d-proline) have also been found in the metabolites of *B. casei* G20. These make Coleoptera culturable endosymbiont bacteria have the industrial potential to distinguish from other saline-tolerant microorganisms.

## Conclusion

The similar composition of endosymbiont bacteria of Coleoptera and its culturable endosymbiont bacteria confirmed the reliability of the screening results, and these culturable endosymbiont bacteria generally had a certain tolerance to saline-alkali environment. Because endosymbiont bacteria participate in almost all the life activities of the host synergistically, it can enhance the adaptability of the host environment. Combined with the analysis of the living areas of the sample insects, the soil salinization in arid and semi-arid climates is highly frequent. In order to cope with the arid climate and saline-alkali environment, the Coleoptera insects in this region have coevolved with their host strains. Some strains can not only produce betaine, amino acids, polyols, and other compounds to enhance the resistance of the host but also produce previously unreported metabolites related to the saline-alkali tolerance mechanism, providing a sample source for the mining of novel functional molecules. It is a necessary basis for the transformation of extremophile "cell factory" to create and enrich the extremophile strain bank by exploring different types of extremophile. Expanding the range of living environment for screening extremophiles is conducive to studying the universal resistance mechanism, so as to provide more accurate and convenient services for the bacterial chassis transformation.

## ACKNOWLEDGMENTS

This work was funded by the Third Xinjiang Scientific Expedition Program, the National Key Research and Development Program of China (grant 2022xjkk020603); the Outstanding Young Scientific and Technological Talents Training Program of Xinjiang Autonomous Region (grant 2020Q02); the National Natural Science Foundation of China (grant U2003305, 31860018); and the Tianshan Innovation Team Project of Xinjiang Autonomous Region (grant 2020D14022).

## AUTHOR AFFILIATIONS

[1]Laboratory of Synthetic Biology, Department of Bioengineering, School of Life Science and Technology, Xinjiang University, Urumqi, China
[2]School of Future Technology, Xinjiang University, Urumqi, China

## AUTHOR ORCIDs

Haitao Yue  http://orcid.org/0000-0001-9429-7631

## FUNDING

| Funder | Grant(s) | Author(s) |
|---|---|---|
| Third Xinjiang Scientific Expedition Program, National Key Research and Development Program of China | 2022xjkk020603 | Haitao Yue |
| Outstanding Young Scientific and Technological Talents Training Program of Xinjiang Autonomous Region | 2020Q02 | Haitao Yue |
| National Natural Science Foundation of China | U2003305 31860018 | Haitao Yue |
| Tianshan Innovation Team Project of Xinjiang Autonomous Region | 2020D14022 | Haitao Yue |

## DATA AVAILABILITY

The sequence of *Brevibacterium casei* G20 has been deposited in GenBank under accession no. PRJNA754761.

## ADDITIONAL FILES

The following material is available online.

Open Peer Review

**PEER REVIEW HISTORY (review-history.pdf).** An accounting of the reviewer comments and feedback.

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
