## [Reviewer comments · Microbiology Spectrum]

Microbiology Spectrum

Diversity and Saline-alkali Resistance of Coleoptera Ensymbiont Bacteria in Arid and Semi-arid Climate

Haitao Yue, Xiaoyun Ma, Shuwen Sun, Hongying Hu, Jieyi Wu, Tong Xu, Danyang Huang, Yiqian Luo, Junqiang Wu, and Tingting Huang

Corresponding Author(s): Haitao Yue, Laboratory of Synthetic Biology

Review Timeline:

Submission Date:	February 6, 2024
Editorial Decision:	March 14, 2024
Revision Received:	March 21, 2024
Accepted:	April 21, 2024

Editor: Jing Han

Reviewer(s): Disclosure of reviewer identity is with reference to reviewer comments included in decision letter(s). The following individuals involved in review of your submission have agreed to reveal their identity: Pralay Shankar Gorai (Reviewer #1); İkbâl Agah İnce (Reviewer #2)

Transaction Report:

DOI: <https://doi.org/10.1128/spectrum.00232-24>

Re: Spectrum00232-24 (Diversity and Saline-alkali Resistance of Coleoptera Endophytic Bacteria in Arid and Semi-arid Climate)

Dear Prof. Haitao Yue:

Thank you for the privilege of reviewing your work. Below you will find my comments, instructions from the Spectrum editorial office, and the reviewer comments.

Revision Guidelines

Sincerely,
Jing Han
Editor
Microbiology Spectrum

Reviewer #1 (Comments for the Author):

Comments and suggestions for the authors

1. You have written Coleoptera endophyte but endophyte take shelter in the plant tissue at least part of its life cycle. So, how can you tell the term 'endophyte' here? Have you found that those bacteria reside in the plant at least part of its life cycle?
2. Have you performed any confirmative test which can assure that strain B. casei G20 is an endosymbiont of Coleoptera

insects?

3. It will be better if you use only strain number instead of genus and species name in the methodology section.

4. Manuscript is well written and in the present scenario identified strain is significant.

Reviewer #2 (Comments for the Author):

Thank you for the authors properly addressing all the comments from the reviewers.

Dear Editor and Reviewers,

We are very grateful for your careful review of the manuscript and for providing a summary and encouraging comments on the manuscript. According to the editor and reviewers' comments, we have made some changes marked in red in revised paper, which will not influence the content and framework of the paper. We appreciate for editors and reviewers hard work and hope that the corrections will meet with approval. Once again, thank you very much for your comments and suggestions.

The reviewer comments are laid out below in italicized font, and specific concerns have been numbered. Our response is given in normal font and changes/additions to the manuscript are given in the blue text.

Reviewer #1 (Comments for the Author):

Comments and suggestions for the authors

① *You have written Coleoptera endophyte but endophyte take shelter in the plant tissue at least part of its life cycle. So, how can you tell the term 'endophyte' here? Have you found that those bacteria reside in the plant at least part of its life cycle?*

The author's response: Thank you for your meticulous scientific advice. Regarding the use of 'endophyte' and 'endophytic', previously we wanted to use them to describe the strains we screened. There is relatively limited research on useful microorganisms in the order Coleoptera and detailed references are scarce. Therefore, we wanted to quote the professional terminology used in the study of plant endophytic bacteria. However, through your advice and some literature search, we found that the

prefix ‘phy’ indicates a connection with plants or something involving plants. This prefix usually comes from the Greek word root ‘phyto’ meaning ‘plant’ . Thus, our usage in this context is indeed inappropriate. Following your suggestion, we have replaced ‘endophyte’ or ‘endophytic’ with ‘endosymbiont’ throughout the entire text.

As for whether the screened strains colonize plants at some point in their life, we believe the following factors may affect their life stages. The research indicates that the Coleoptera insects often colonize plants during the larval stage. After a long period of coevolution with the plants and their endosymbionts, it has been found that the presence of plant DNA can also be detected in the intestines of the insects as they grow into their adult stage (Jacqueline et al., 2024). This may imply a certain relationship between the microorganisms isolated from Coleoptera and plants. However, because our study focuses on exploring the coevolutionary relationship between hosts (Coleoptera) and endosymbionts in saline-alkali environments and some available resources, we did not extensively discuss the life habits of Coleoptera in its various growth stages. Of course, we are continuing to delve into the connection between insects and their microorganisms in the hope of discovering more intriguing scientific phenomena.

② *Have you performed any confirmative test which can assure that strain B. casei G20 is an endosymbiont of Coleoptera insects?*

The author's response: Thank you for your constructive feedback. Based on this scientific evidence, we conducted a microbial diversity analysis on the insect bodies.

When we collected samples, these insects were in their natural living state. During transportation, storage, and experimental operation processes, we minimized human experimental errors to the greatest extent possible.

Subsequently, we conducted microbial diversity testing on the insect bodies, and the diversity data results of each sample are consistent with the screening results, mutually verifying each other. According to the research method of gut content analysis of a phloem-feeding insect (Cooper et al., 2016), we conducted surface cleaning of the insect bodies by soaking them in a 75% ethanol solution for 30 seconds. After removing the insect bodies, they were soaked in a 1% sodium hypochlorite solution for 5 minutes and rinsed 10 times with sterile water. To assess the effectiveness of surface disinfection, the sterile water from the final rinse was spread on LB solid medium and incubated at 37°C for two days. If no colonies appeared on the plate, it indicated thorough sterilization of the insect body surfaces.

During the screening of strains, we dissected the insect bodies using the most rigorous operational methods to ensure the scientific rigor of the screening process and to avoid any potential errors caused by the external environment. The insects we studied were all in the adult stage. We believe that studying insects in their adult stage can better illustrate the evolutionary relationship between internal microorganisms and insect bodies, thus addressing scientific questions.

Therefore, we can only confirm that the strains originate from inside the insects, as we did not track the life history of the insects, and we did not conduct in-depth research on specific tissue locations.

③ *It will be better if you use only strain number instead of genus and species name in the methodology section.*

The author's response:We sincerely thank the reviewer for careful reading. As suggested by the reviewer, we have corrected the '*Brevibacterium casei* G20 'into '*Brevibacterium linens* YS ' .(Line195-196,Line210-21 and Line226)

④ *Manuscript is well written and in the present scenario identified strain is significant.*

The author's response:Thank you for the encouragement from the reviewers. We take all the feedback seriously and look forward to making new discoveries in our ongoing research.

Reviewer #2 (Comments for the Author):

① *Thank you for the authors properly addressing all the comments from the reviewers.*

The author's response:We sincerely thank the reviewers for their careful reading and words of encouragement, and we look forward to our work being recognized.

Re: Spectrum00232-24R1 (Diversity and Saline-alkali Resistance of Coleoptera Endosymbiont Bacteria in Arid and Semi-arid Climate)

Dear Dr. Haitao Yue:

Your manuscript has been accepted, and I am forwarding it to the ASM production staff for publication. Your paper will first be checked to make sure all elements meet the technical requirements. ASM staff will contact you if anything needs to be revised before copyediting and production can begin. Otherwise, you will be notified when your proofs are ready to be viewed.

Sincerely,
Jing Han
Editor
Microbiology Spectrum

Reviewer #2 (Comments for the Author):

I recently had the opportunity to review your abstract titled "Diversity and Saline-alkali Resistance of Coleoptera Endosymbiont Bacteria in Arid and Semi-arid Climate" and I must commend you on the intriguing insights you have provided regarding the endosymbiotic microorganisms of Coleoptera in arid and saline environments. Your study sheds light on an underexplored area of research and offers valuable implications for understanding microbial adaptation to extreme conditions.

While I found your findings to be highly compelling, I believe that there is significant potential for enhancing the impact of your study through further refinement of the data analysis section. Specifically, I would like to suggest some areas for major improvement:

While you have presented results regarding the abundance and functional predictions of endosymbiont bacteria in Coleoptera, a more comprehensive statistical analysis would greatly strengthen the robustness of your findings. Incorporating appropriate statistical tests to assess the significance of observed differences and correlations would provide greater confidence in the reliability of your conclusions.

The differential metabolite and protein analysis of *Brevibacterium casei* G20 during different growth phases offer valuable insights into its adaptive responses to saline environments. However, to fully leverage the potential of multi-omics data, I

recommend integrating these findings with other omics datasets (e.g., transcriptomics, metagenomics) to gain a more holistic understanding of microbial adaptation mechanisms.

While you have highlighted the presence of inorganic ion transporters and metabolism-related pathways in the endosymbiont strains, further elaboration on the functional significance of these findings would enhance the interpretability of your results.

Providing insights into how these metabolic pathways contribute to salinity tolerance and microbial adaptation would enrich the discussion and broaden the impact of your study.

Including a discussion of methodological considerations, such as sample processing techniques, quality control measures, and potential limitations of the analytical methods employed, would provide important context for interpreting the results and help readers assess the robustness of the findings.

Incorporating these suggestions into the data analysis section of your study has the potential to significantly enhance its scientific rigor and impact. I am confident that with these improvements, your study will make an even greater contribution to the field of microbial ecology and adaptation to extreme environments.